# Practices for preventing Hepatitis B infection among health science students in Ethiopia: Systematic review and meta-analysis

Gemeda Wakgari Kitil[1]*, Abiy Tasew Dubale[2], Adamu Ambachew Shibabaw[2], Alex Ayenew Chereka[2]

1 Departments of Midwifery, College of Health Sciences, Mattu University, Mattu, Ethiopia, 2 Departments of Health Informatics, College of Health Sciences, Mattu University, Mattu, Ethiopia

* gemedawa425@gmail.com

**Data Availability Statement:** All data utilized and examined in this study are openly accessible within the supporting files.

## Abstract

### Background

Hepatitis B virus infection remains a significant public health concern globally, particularly among healthcare workers, including health science students who are at high risk due to their exposure to infected patients and contaminated medical equipment. In Ethiopia, where the burden of HBV infection is substantial, preventive practices among health science students are critical for minimizing transmission and ensuring a healthy workforce. However, there is a lack of comprehensive evidence regarding the effectiveness of these practices specifically among this population in Ethiopia. Therefore, this study aimed to provide a systematic review and meta-analysis of preventive measures for Hepatitis B infection among Health Science Students in Ethiopia.

### Methods

This study followed the guidelines outlined in the PRISMA checklist and focused on research conducted within Ethiopia. Seven relevant studies were identified through comprehensive searches across various databases including Google, Medline, PubMed, and Scholar. Data retrieval was systematically conducted using a checklist, and analysis was performed using STATA version 14. Heterogeneity was assessed using both the Cochrane Q test and the $I^2$ statistic. Additionally, publication bias was evaluated using Egger's weighted regression, a funnel plot, and Begg's test.

### Results

In this meta-analysis and systematic review, we identified a total of 515 research articles, of which seven studies met the eligibility criteria for analysis. The overall pooled magnitude of practices aimed at preventing Hepatitis B infection among Health Science Students in Ethiopia was 41.21% (95% CI: 30.81–51.62). Factors significantly associated with these practices included better understanding of Hepatitis B infection prevention (OR = 1.99, 95% CI: 1.20–3.29), age group 20–24 years (OR = 5.79, 95% CI: 2.43–13.78), needle stick injury

**Funding:** The author(s) received no specific funding for this work.

**Competing interests:** The authors confirm the absence of any known financial or personal conflicts of interest that could have influenced the study's findings

exposure (OR = 3.43, 95% CI: 1.10–10.70), and students enrolled in medicine or public health officer departments (OR = 4.20, 95% CI: 2.65–6.65).

## Conclusion

Our analysis indicates that only 41.21% of Health Science students in Ethiopia adhere to Hepatitis B prevention practices. To improve these practices, it is essential to mandate vaccination, provide targeted training on infection prevention, and increase awareness of vaccine uptake. Tailored educational programs should equip students with practical strategies. Additionally, intelligent interventions must address factors influencing preventive practices. Collaboration between institutions and ongoing monitoring is crucial to ensuring success.

## 1. Introduction

Hepatitis B virus infection poses a significant worldwide public health challenge, as demonstrated by a study revealing that 3.61% of the total burden of HBV infection is observed globally, with notably higher burdens recorded in the western Pacific region (5.26%) and African nations (8.8%) [1]. Based on WHO estimates for 2024, approximately 254 million individuals were identified as having chronic hepatitis B infection in 2022, with 1.2 million new infections occurring each year. This resulted in an estimated 1.1 million deaths, primarily due to cirrhosis and hepatocellular carcinoma (primary liver cancer) [2]. Around 350 million individuals globally are thought to be chronically infected with the hepatitis B virus (HBV), out of an estimated two billion total HBV infections [3].

A study indicates that the infection burden ranges from 6% to 9%, making it the most frequently transmitted blood-borne disease in hospital settings. Healthcare professionals and medical students, due to their clinical affiliations that expose them to blood and bodily fluids, are at a higher risk of contracting HBV [4, 5]. Individuals working in healthcare are at a heightened risk of contracting HBV through mucosal exposures, such as contact with non-intact skin, improperly sterilized equipment, and exposure to infected blood or bodily fluids. Additionally, percutaneous injuries like needle sticks and sharp object injuries increase this risk. Healthcare workers are four times more likely to become infected with HBV compared to the general population [6]. Medical and health science students are highly susceptible to contracting the hepatitis B virus and transmitting it to others due to their involvement in clinical practice and exposure to blood or other bodily fluids in laboratories, public safety settings, and patient care [7].

Several epidemiological studies have been conducted to assess the prevalence of good practices for preventing HBV among medical and health science students. These studies have revealed a wide range of magnitude rates. For instance, in Nepal, the magnitude stands at 14.2% [8] while in India, it is significantly higher at 76.59% [9]. Similarly, in Ethiopia, the magnitude varies from 30.2% in the Amhara region [10] to 50.3% in Southern Ethiopia [11]

Numerous factors influence preventive practices regarding Hepatitis B infection among health science students. These factors include their level of knowledge, attitude, age, residence, marital status, and their respective departments, which encompass medicine and public health. Additionally, exposure to needle stick injuries plays a significant role in shaping preventive behaviors [11–15].

Preventing Hepatitis B virus (HBV) infection entails adhering to universal precautions, which involve utilizing protective measures like gloves, ensuring proper sterilization of medical equipment, managing hospital waste effectively, and implementing vaccination protocols.

Furthermore, in cases of accidental exposure to contaminated blood or bodily fluids, post-exposure prophylaxis (PEP) can serve as an additional preventive measure against HBV transmission [16–18].

The World Health Organization (WHO) has set a target to eliminate the Hepatitis B Virus (HBV) by the year 2030 [19]. However, despite this ambitious goal, preventive measures aimed at combating Hepatitis B infection among health science students continue to be insufficient in numerous developing nations, Ethiopia included [20].

In Ethiopia, there are several studies on preventive practices against Hepatitis B infection. However, there's a gap in having a comprehensive national estimate and understanding of associated factors among Health Science Students. Previous research has provided inconclusive, varied, and regionally limited findings. Since Hepatitis B virus (HBV) infection remains a significant public health concern worldwide, particularly among healthcare professionals, including health science students In Ethiopia, where the burden of HBV infection is considerable, preventative measures among health science students are vital to curb transmission rates and protect both students and patients. This systematic review and meta-analysis aim to comprehensively evaluate a nationally representative overview of preventive practices towards Hepatitis B infection and their associated factors in Ethiopia. By synthesizing existing evidence, this study seeks to identify effective preventive strategies, gaps in current practices, and areas for improvement. Understanding the landscape of HBV prevention among health science students is essential for informing targeted interventions, enhancing educational initiatives, and ultimately reducing the burden of HBV infection in Ethiopia's healthcare settings.

## 2. Materials and methods

### 2.1 Source of information and search strategy

In our search for pertinent research, we meticulously explored both the PROSPERO database and the Database of Abstracts of Reviews of Effects (DARE), accessible at http://www.library. UCSF.edu. Following the guidelines set forth by the Preferred Reporting Items for Systematic Reviews and Meta-Analysis (PRISMA) [21], our study aims to explore factors closely associated with the practices aimed at preventing Hepatitis B infection among Health Science Students in Ethiopia (S1 Table).

We systematically searched multiple online databases, including Google Scholar, HINARI, EMBASE, PubMed, Web of Science, the African Journal Online (AJOL), Global Health, and Scopus, from November 3, 2023, to February 13, 2024, to conduct a thorough literature review. Our search method utilized a combination of keywords, free-text search queries, and Medical Subject Headings (MeSH). We employed Boolean operators to combine various terms related to "practices," magnitude," "associated factors," "determinants," and "prevention of Hepatitis B infection" with terms specifically related to health science students. The structured search criteria were as follows: ("assessment," "practices," magnitude"," associated factors," "determinants," "prevention of Hepatitis B infection," "practices toward prevention of hepatitis B virus infection," "Health Science students," "medicine and health sciences Students") AND ("prevention of Hepatitis B infection," OR "practices toward prevention of hepatitis B virus infection" OR "practices toward prevention of hepatitis B virus infection of Health Science students" AND "Ethiopia"). This methodical approach guaranteed an extensive and systematically structured examination of the literature within the designated period.

The primary aim of this study was to thoroughly investigate the prevention measures employed by Health Science Students in Ethiopia to combat Hepatitis B infection and associated factors. We employed a meticulous research methodology, examining titles, abstracts, and full texts of selected studies. To make sure we got the data right, we followed a standard

method recommended by the Joanna Briggs Institute [22]. Three reviewers, GWK, AAS, and AAC, worked independently to extract the information from each article. If we found any differences, we discussed them together to make sure our findings were reliable.

This systematic review and meta-analysis form the basis for a thorough examination of the practices of Health Science Students in Ethiopia regarding the prevention of Hepatitis B infection. The study aims to provide a comprehensive understanding of the literature and to establish a framework for gaining valuable insights into the factors influencing the practices of Health Science Students in preventing Hepatitis B infection in Ethiopia.

## 2.2 Eligibility criteria

In this systematic review, our inclusion criteria were structured to target original research studies examining the practices of Health Science Students in preventing Hepatitis B infection and associated variables within the Ethiopian context. We included observational studies, specifically cross-sectional, cohort, and case-control studies that reported on preventive practices and measures against Hepatitis B infection among health science students in Ethiopia. By not imposing restrictions based on publication year, we aimed to ensure a comprehensive and up-to-date synthesis of the available literature.

The review covered publications until February 13, 2024, presenting an up-to-date view of research. To ensure accuracy and relevance, a strict exclusion criterion was enforced: studies not disclosing Health Science Students' practices in preventing Hepatitis B infection in Ethiopia were disregarded. Additionally, articles lacking abstracts or full texts were excluded to facilitate a thorough examination of selected studies. This meticulous approach aimed to maintain the quality and applicability of the literature analyzed in the study.

The primary focus of this systematic review was to examine cross-sectional and analytical research on the practices of Health Science Students in preventing Hepatitis B infection in Ethiopia. We specifically included full-text English papers from peer-reviewed journals that were readily accessible. Conversely, non-cross-sectional research, such as case reports, conference reports, national survey reports, and expert comments, as well as studies conducted in languages other than English, were excluded. Additionally, editorial reports, reviews, letters, or commentary were not considered in this comprehensive evaluation.

This systematic review and meta-analysis adhered to the PICO framework, focusing on the population of health science students in Ethiopia (P). The intervention of interest involved various practices aimed at preventing Hepatitis B infection among these students, including vaccination, utilization of personal protective equipment, and adherence to infection control protocols (I). By comparing different approaches or levels of adherence to these preventive measures (C), the study aims to evaluate their efficacy in reducing the incidence or transmission of Hepatitis B infection among health science students in Ethiopia (O).

## 2.3 Quality assessment and appraisal

To determine the quality of the studies, we used a standardized tool to identify potential biases and understand any differences in the research findings. Two independent reviewers conducted a quality control check. We evaluated methodological issues using the Newcastle-Ottawa Scale (NOS), specifically designed to assess bias in observational research. Publications deemed relevant were those scoring seven or higher on the modified NOS scale (S2 Table).

## 2.4 Data extraction process

The study's quality was evaluated using the Joanna Briggs Institute (JBI) quality assessment checklist, specifically designed for cross-sectional studies [22]. Data extraction was

accomplished by two authors (AAS and AAC) using a predefined checklist in Microsoft Excel to gather pertinent information (**S3 Table**) including details such as the authors' names, publication year, study area, study design, sample size, magnitude, response rate, and the number of Health Science students involved.

To streamline the process, the initial steps involved consolidating search results from multiple databases and utilizing reference management software (EndNote version 20.0) to identify and remove duplicate articles. Subsequently, a thorough review of study titles and abstracts was conducted to exclude irrelevant entries. The assessment of full-text publications formed the basis for a comprehensive evaluation of the remaining research articles. In cases of disagreement between reviewers AAC and AAS, we resolve them through additional feedback and consensus-building discussions. If the main articles lack sufficient information, we reach out to the corresponding authors via email for clarification.

### 2.5 Outcome measurement

There were two main outcomes in this study. The primary outcome was to estimate the overall magnitude of preventive practices among Health Science students toward Hepatitis B infection. The secondary outcome involved analyzing the collective factors associated with these preventive practices.

### 2.6 Data synthesis and analysis

Data extraction for the synthesis and analysis process was conducted using a Microsoft Excel spreadsheet. Subsequently, the extracted data were inputted into STATA version 14 for further examination. Primary studies were comprehensively described and summarized through the use of tables, forest plots, and figures. To derive an overall estimate of practices for preventing Hepatitis B infection, a random-effects model with a 95% confidence interval (CI) was applied.

To evaluate the relationship between factors relevant to Health Science Students, we used odds ratios with 95% confidence intervals. Due to the observed variability among the studies included, we chose a random-effects model for the meta-analysis. This decision was made to ensure a thorough analysis and interpretation of the compiled data.

To understand how much variation there was in reported magnitude among studies, we used $I^2$ statistics and Cochran's Q test. If the p-value from Cochran's Q test was less than 0.05, we considered it statistically significant. The $I^2$ statistic, which ranges from 0% to 100%, tells us the level of variation: no variation (0%), a little (25%), moderate (50%), or high (75%). To check if there might be bias in what studies get published, we also did Egger regression tests and looked at the shape of the funnel plot.

### 2.7 Ethical approval and consent for publication

This study strictly adheres to the Preferred Reporting Items for Systematic Reviews and Meta-Analysis (PRISMA) guidelines, which ensure transparency and rigor in reporting, ethical clearance was not deemed necessary. The primary focus of this research is to synthesize and analyze existing data, rather than directly involving human subjects or primary data collection. Therefore, ethical clearance was not sought nor required for this study.

## 3. Results

### 3.1 Study selections

During the initial phase of our study, we gathered 515 articles from various databases. After a rigorous screening process to remove duplicate articles, we excluded 100. Subsequently, we

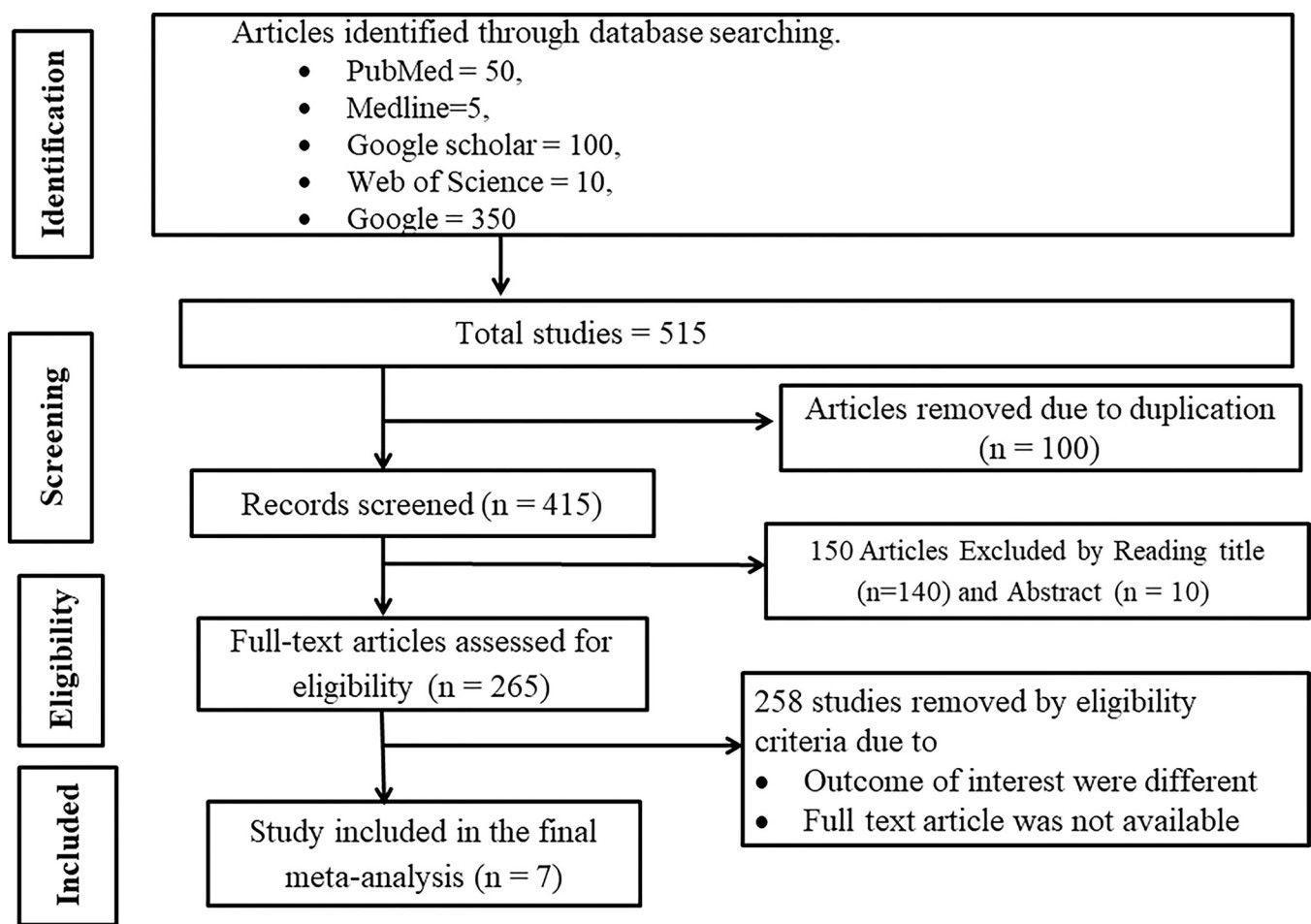

**Fig 1. A diagram illustrating the PRISMA method for the selection of articles in systematic review and meta-analysis 2024.**

further refined our selection based on a thorough assessment of abstracts and titles, eliminating an additional 150 articles. Following this screening process, we conducted a detailed full-text review of the remaining 265 articles. Finally, in the last stage of our meta-analysis and systematic review, we meticulously examined the seven selected research articles for inclusion (**Fig 1**).

### 3.2 Characteristics of included studies for review

In our comprehensive meta-analysis and systematic review, we synthesized findings from seven studies conducted in Ethiopia. Notably, the study carried out in the Amhara region [14, 15] had the largest participant pool, totaling 422 individuals, whereas the study in the same region [23] had the smallest, with 200 participants. Overall, the combined sample across all studies consisted of 2,433 care providers, varying from 200 to 422 participants per study. It's worth noting that each study employed a cross-sectional design.

In terms of the study locations, four studies took place in the Amhara region [10, 14, 15, 23] two were conducted in SNNPR [11, 24], and one study was carried out in Oromia [19] (**Table** 1).

### 3.3 Pooled magnitude of the preventive practice of Hepatitis B infection

The overall pooled magnitude of practices aimed at preventing Hepatitis B infection among Health Science Students in Ethiopia was 41.21% (95% CI: 30.81–51.62). Among the

**Table 1. A descriptive summary of seven studies included in the meta-analysis on practices of preventing Hepatitis B infection among health science students in Ethiopia, 2024.**

| Authors | Year | Region | Study area | Study design | Study population | Quality score | Response rate | Sample Size | Practice |
|---|---|---|---|---|---|---|---|---|---|
| Allene et al [15] | 2020 | Amhara | Debre Berhan University | cross-sectional | Health Science students | 9 | 84.12% | 422 | 33.8% |
| Abdela et al [10] | 2016 | Amhara | University of Gondar. | cross-sectional | Health Science students | 8 | 100% | 246 | 30.2% |
| Aynalem et al [11] | 2022 | SNNPR | Hawassa University | cross-sectional | Health Science students | 8 | 98% | 404 | 50.3% |
| Gebremeskel et al [23] | 2019 | Amhara | Woldia University | cross-sectional | Health Science students | 9 | 100% | 200 | 39.5% |
| Mesfin et al [19] | 2013 | Oromia | Haramaya University | cross-sectional | Health Science students | 9 | 100% | 322 | 40.68% |
| Demsis et al [14] | 2018 | Amhara | Wollo University | cross-sectional | Health Science students | 7 | 96.70% | 422 | 50.0% |
| Haile et al [24] | 2021 | SNNPR | Wolkite University | cross-sectional | Health Science students | 9 | 100% | 417 | 47.0% |

included studies, the study conducted at Hawasa University in the southern region reported the highest magnitude at 50.30% (95% CI: 22.01–78.59), while the University of Gondar in the Amhara region recorded the lowest magnitude at 30.20% (95% CI: 4.22–56.18) [10, 11]. The analysis, with an $I^2$ value of 0.0% and a p-value of 0.926, revealed no substantial heterogeneity among the included studies, indicating that subgroup analysis was unnecessary (**Fig 2**).

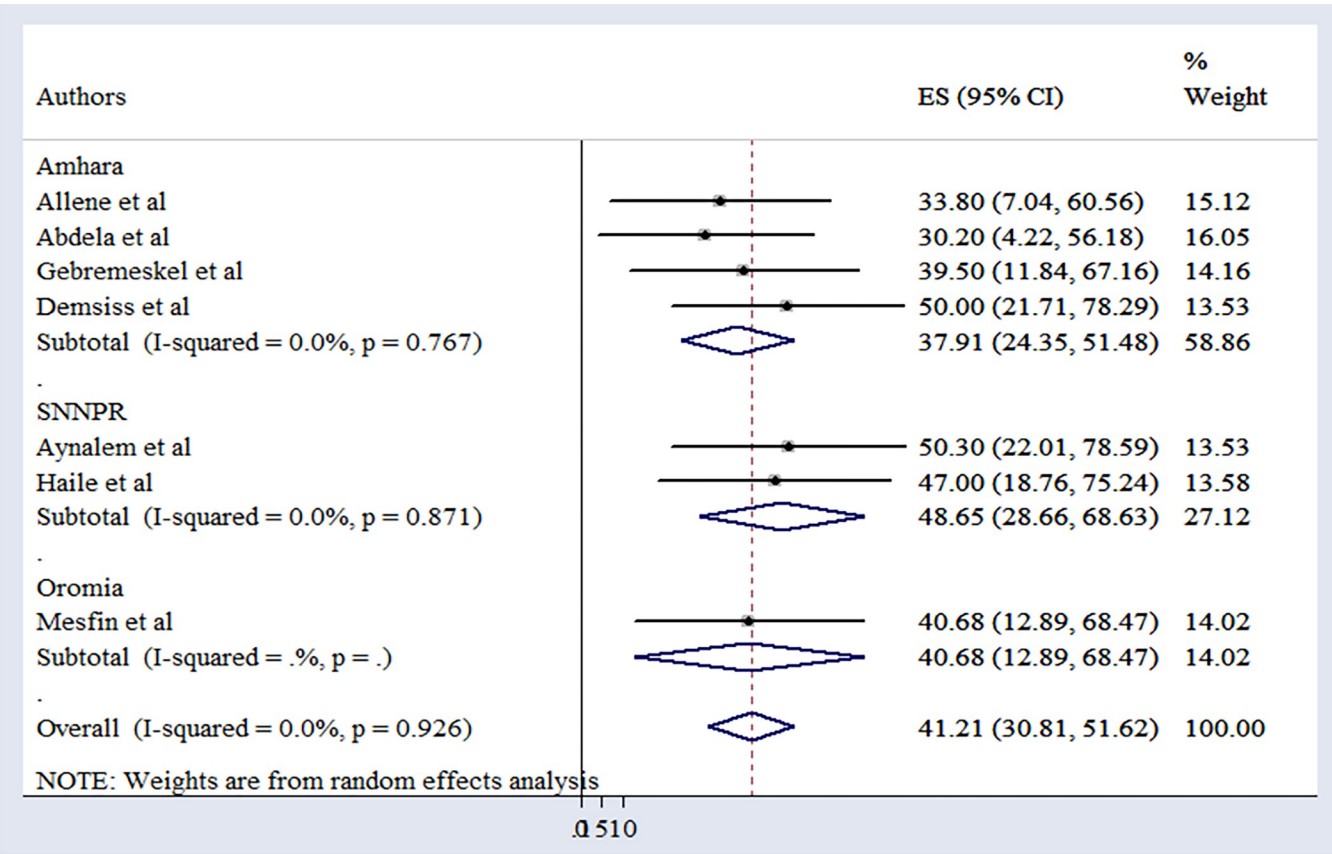

**Fig 2. Forest plot depicting the pooled practices for preventing Hepatitis B infection in Ethiopia.**

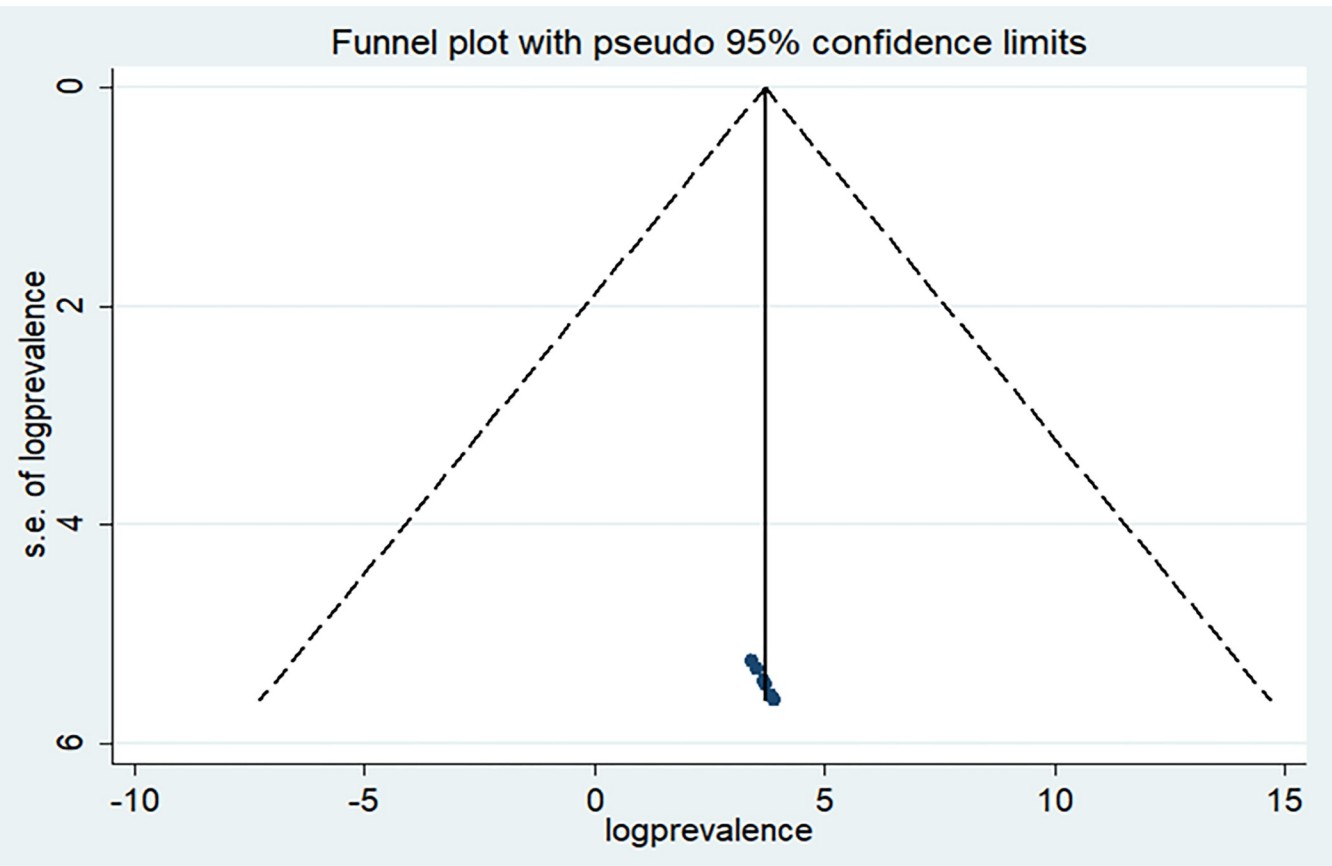

**Fig 3. Funnel plot of included studies to test publications bias.**

### 3.4 Publications bias

We utilized both a visual assessment of a funnel plot and Egger's regression test to examine publication bias. Although the funnel plot displayed an uneven distribution upon initial observation, the outcome of Egger's test (P = 0.35) did not achieve statistical significance. Consequently, based on the findings of Egger's regression test, there is no substantiated evidence suggesting the presence of publication bias within the research articles analyzed in our meta-analysis (**Fig 3**).

### 3.5 Factors associated with preventive practices aimed at Hepatitis B infection

In assessing the factors collectively influencing preventive practices targeting Hepatitis B infection, we conducted a meta-analysis of seven studies. Using the command "metan logor selogor, xlab(0.1, 1, 10) label(namevar = authors) by (factors) random texts(180) eform," we evaluated the cumulative effects of the odds ratio.

From our analysis exploring the determinants of preventive practices aimed at Hepatitis B infection, several key factors emerged. One significant factor associated with the adoption of preventive practices is knowledge. Our analysis indicates that students with a better understanding of Hepatitis B infection prevention are almost twice as likely to practice HBV infection prevention effectively compared to those with lower levels of knowledge (OR = 1.99, 95% CI: 1.20–3.29).

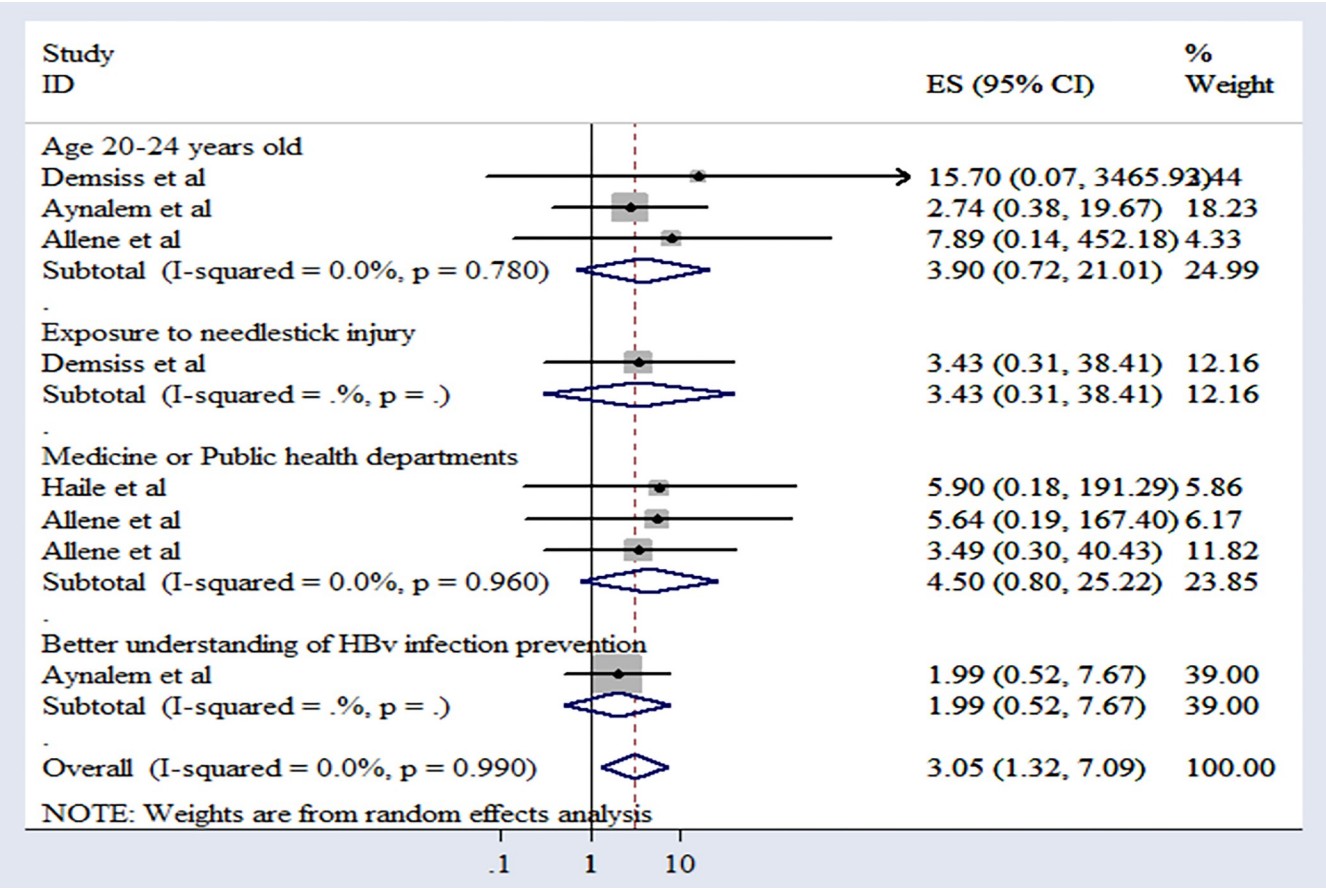

**Fig 4. Forest plot of associated factors of practice towards prevention of Hepatitis B infection in Ethiopia, 2024.**

Exposure to needle stick injuries emerged as another significant factor influencing preventive practices. Our analysis revealed that students who have experienced such incidents are significantly more likely to adopt preventive measures (OR = 3.43, 95% CI: 1.10–10.70).

Moreover, our study showed that students aged 20 to 24 (OR = 5.79, 95% CI: 2.43–13.78) were more likely to have good practice towards hepatitis B virus infection prevention.

Finally, the department of study emerges as a significant predictor of preventive practices, with students from medicine or public health officer departments demonstrating higher adherence rates compared to others (OR = 4.20, 95% CI: 2.65–6.65) (**Fig 4**).

## 4. Discussion

The findings from our systematic review and meta-analysis revealed a concerning situation regarding the magnitude of preventive practices aimed at Hepatitis B infection among Health Science students in Ethiopia. With only 41.21% (95% CI: 30.81–51.62) of students demonstrating adherence to recommended preventive measures, there is a clear need for improvement in infection control strategies within educational settings.

While a meta-analysis for this field of research in Ethiopia is not yet available, our study's pooled magnitude of preventive practices targeted at Hepatitis B infection aligns with the findings of a previous study among Saudi Arabian medical and health science college students [25]. That study indicated that 47.2% of the participants demonstrated good practices toward

HBV infection prevention. The possible justification for this similarity is that both studies target college students in the medical and health science fields. Since the participants come from similar educational backgrounds, there may be commonalities in their knowledge, awareness, and access to resources related to Hepatitis B infection prevention.

The results of this study, however, are higher than those of the research done among clinical-year medical students at a medical college in Nepal [8], which revealed that just 14.2% of the study respondents had good practice. This disparity could arise from variations in study participants' perspectives on HBV prevention, differences in educational curricula, discrepancies in clinical practice, the sociocultural context of the study area, and the accessibility of resources.

Our study findings reveal that participants' levels of good practice are lower compared to those reported in similar studies. Specifically, among medical students at Tanta University in Egypt [26], 68.1% exhibited good practice, while in a separate study among medical students at a tertiary care center in Tamil Nadu, India, the figure was even higher at 76.59% [9]. This disparity could stem from variations in training methods, adherence to infection prevention and control guidelines among students, differences in educational backgrounds, and variations in the epidemiology of the hepatitis B virus in the regions under study. Furthermore, the insufficient facilities and equipment in the research location underscore the need to rectify shortcomings by enhancing education on general safety measures and allocating resources appropriately.

In our analysis investigating the determinants of preventive practices against Hepatitis B infection, several key factors emerged. Notably, knowledge emerged as a significant factor associated with the adoption of preventive practices. Our analysis indicates that students with a better understanding of Hepatitis B infection prevention are nearly twice as likely to engage in these practices compared to those with lower levels of knowledge (OR = 1.99, 95% CI: 1.20–3.29). This result is in line with a study conducted by University Putra Malaysia's international students [27]. This may be because an increase in students' knowledge of HBV prevention may lead to a corresponding increase in their practice of preventing hepatitis B virus infection.

Exposure to needle stick injuries emerged as another significant factor influencing preventive practices. Our analysis revealed that students who have experienced such incidents are significantly more likely to adopt preventive measures (OR = 3.43, 95% CI: 1.10–10.70). This finding is consistent with a study carried out among medical students in Cameroon [12].

Furthermore, our study revealed a significant trend: students aged between 20 and 24 showed a higher likelihood of adhering to practices aimed at preventing hepatitis B virus infection (OR = 5.79, 95% CI: 2.43–13.78). This inclination could be attributed to the magnitude of this age group among graduating students across various departments, suggesting a pivotal stage in education where awareness campaigns and educational interventions could be particularly effective in promoting preventive behaviors against hepatitis B virus infection.

Finally, our study reveals that the department of study plays a crucial role in predicting preventive practices, particularly evident among students enrolled in medicine or public health officer departments, who exhibit notably higher adherence rates (OR = 4.20, 95% CI: 2.65–6.65). This observation aligns with findings from a study conducted in China [13], corroborating the importance of departmental influence on preventive behaviors. This trend may be explained by the significantly higher adherence rates to preventive practices among students enrolled in medicine or public health officer departments. Several factors contribute to this, including a specialized curriculum that emphasizes preventive measures, hands-on experience gained through internships, a professional culture that prioritizes prevention, access to resources and information, and mentorship from faculty members. Together, these factors

cultivate a deeper understanding and commitment to preventive healthcare among students in these departments.

## 5. Conclusion and recommendation

### 5.1 Conclusion

In conclusion, our study states that preventive practice is low among Health Science students in Ethiopia, with only 41.21% demonstrating adequate adherence to Hepatitis B prevention measures. Factors influencing adherence include a thorough understanding of Hepatitis B prevention, belonging to the 20–24 age group, experiencing needle stick injuries, and being enrolled in medicine or public health officer departments.

### 5.2 Recommendation

Implement comprehensive educational programs tailored for Health Science students, focusing on Hepatitis B prevention. Emphasize the importance of vaccination, infection control measures, and understanding the risks associated with needle stick injuries. These programs will help increase awareness and knowledge, leading to better preventive practices.

Provide regular training sessions on safe practices, including the proper handling of needles and sharps, to reduce the incidence of needle stick injuries among students.

Integrate Hepatitis B prevention education into the core curriculum of medicine and public health officer departments. Ensure that students receive continuous education on this topic throughout their academic careers. By making this a part of their regular studies, students will have a stronger and more consistent understanding of prevention strategies.

Establish support services for students who experience needle stick injuries. These services should include timely medical evaluation, counseling, and follow-up care. Providing these resources ensures that students receive the necessary support and care, minimizing the impact of such injuries.

Develop and enforce institutional policies mandating Hepatitis B vaccination for all Health Science students. Promote strict adherence to infection control guidelines to enhance preventive measures. These policies will help create a safer educational environment and reduce the risk of Hepatitis B transmission.

## Strengths and limitations of the study

Our systematic review and meta-analysis on preventive practices for Hepatitis B infection among Health Science students in Ethiopia provide a robust examination of existing literature, offering clear quantitative estimates crucial for public health strategies. However, limitations include a lack of high-quality studies and potential biases in study selection and generalizability due to predominantly facility-based, cross-sectional studies. Nonetheless, our study provides valuable insights for guiding future research and public health efforts in Ethiopia.

## Supporting information

**S1 Table. PRISMA 2020 checklists of systematic review and meta-analysis of practices for preventing Hepatitis B infection among health science students in Ethiopia.**
(DOCX)

**S2 Table. Quality assessment of systematic review and meta-analysis of practices for preventing Hepatitis B infection among health science students in Ethiopia.**
(DOCX)

**S3 Table. JBI-checklist for systematic reviews and research syntheses.**
(DOCX)

**S1 File. Data set for practice.**
(CSV)

**S2 File. Data set for factors.**
(CSV)

## Acknowledgments

The authors express gratitude to the contributors of the original studies included in this meta-analysis and systematic review, whose work served as the primary source of information.

## Author Contributions

**Conceptualization:** Gemeda Wakgari Kitil, Abiy Tasew Dubale, Adamu Ambachew Shibabaw, Alex Ayenew Chereka.

**Data curation:** Gemeda Wakgari Kitil, Abiy Tasew Dubale, Adamu Ambachew Shibabaw, Alex Ayenew Chereka.

**Formal analysis:** Gemeda Wakgari Kitil.

**Funding acquisition:** Gemeda Wakgari Kitil.

**Investigation:** Gemeda Wakgari Kitil, Abiy Tasew Dubale.

**Methodology:** Gemeda Wakgari Kitil, Alex Ayenew Chereka.

**Project administration:** Gemeda Wakgari Kitil.

**Resources:** Gemeda Wakgari Kitil.

**Software:** Gemeda Wakgari Kitil.

**Supervision:** Gemeda Wakgari Kitil, Abiy Tasew Dubale, Adamu Ambachew Shibabaw.

**Validation:** Gemeda Wakgari Kitil, Alex Ayenew Chereka.

**Visualization:** Gemeda Wakgari Kitil, Abiy Tasew Dubale, Adamu Ambachew Shibabaw.

**Writing – original draft:** Gemeda Wakgari Kitil, Alex Ayenew Chereka.

**Writing – review & editing:** Gemeda Wakgari Kitil, Adamu Ambachew Shibabaw, Alex Ayenew Chereka.

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
