## [Decision Letter · Decision Letter 0]

23 May 2024

PONE-D-24-11778Systematic Review and Meta-Analysis of Practices for Preventing Hepatitis B Infection among Health Science Students in EthiopiaPLOS ONE

Dear Dr. Kitil,

Thank you for submitting your manuscript to PLOS ONE. After careful consideration, we feel that it has merit but does not fully meet PLOS ONE’s publication criteria as it currently stands. Therefore, we invite you to submit a revised version of the manuscript that addresses the points raised during the review process.

We look forward to receiving your revised manuscript.

Kind regards,

Tebelay Dilnessa, MSc

Academic Editor

PLOS ONE

Additional Editor Comments:

Kitil., *et al, *assessed the practices of health science students for preventing hepatitis B infection in Ethiopia. The way presented is very poor and requires a major revision grammatically, typographically, etc.The title "Systematic Review and Meta-Analysis of Practices for Preventing Hepatitis B Infection among Health Science Students in Ethiopia" It is better written, ‘Practices for Preventing Hepatitis B Infection among Health Science Students in Ethiopia: Systematic Review and Meta-Analysis’In the abstract part of conclusion, ‘Factors such as knowledge, age, needle stick injury exposure, and department influence these practices significantly.’ It does not give sense and needs revision. Again ‘ To improve prevention efforts, targeted interventions focusing on education and awareness are crucial. Addressing these disparities can lead to better infection control and public health outcomes in Ethiopia.’ It is a general concept and not emanated from the review result, so needs revision.In the result part of abstract and onwards, please replace the word ‘prevalence’ by ‘magnitude.’In the result part of abstract ‘The statement ‘Factors significantly associated with these practices included good knowledge…..’ It was vague and requires revision.In the methods part and the result part, your citations of figures were not correct. For example, (See S2 Table), (See Error! Reference source not found.), (See 9 Table 1), (see Figure 2), etc. Here, the word ‘see’ was not necessary.Have you taken an ethical clearance? From whom? Is it necessary?The conclusion and recommendation were not targeted and specific.You have to follow the manuscript writing protocol of Plose OneConsidering that the author will significantly revise the paper, major revision was recommended.

Reviewers' comments:

Reviewer's Responses to Questions

**Comments to the Author**

1. Is the manuscript technically sound, and do the data support the conclusions?

Reviewer #1: No

Reviewer #2: Yes

2. Has the statistical analysis been performed appropriately and rigorously? 

Reviewer #1: No

Reviewer #2: Yes

3. Have the authors made all data underlying the findings in their manuscript fully available?

Reviewer #1: Yes

Reviewer #2: Yes

4. Is the manuscript presented in an intelligible fashion and written in standard English?

Reviewer #1: No

Reviewer #2: No

5. Review Comments to the Author

Reviewer #1: 1. The writing lacks fluency and has serious issues with grammar that need to be addressed.

2. The document suffers from major coherence problems and contains redundant ideas.

3. The article’s searching strategy (search string) is inaccurate. The authors failed to identify the key terms related to the PICO (Population, Intervention, Comparison, Outcome) for the study. Consequently, the search string yielded an excessively large number of articles (22,150 articles), which is incorrect. With a properly structured search string, the number of articles should have been limited to several hundred. One of the main reasons for this oversight is the failure to include the country name “Ethiopia” in the search string, despite the study being conducted in Ethiopia. The authors should rectify this issue by restricting the search string with the country name.

4. On page 12, the authors stated, “The analysis, with an I2 value of 0.0% and a p-value of 0.926, revealed no substantial heterogeneity among the included studies.” However, the analysis conducted is not appropriate as it appears to be a subgroup analysis rather than a simple forest plot. Furthermore, even if the analysis was appropriate, it is unclear why the authors conducted further subgroup analysis (figure 4). Additionally, conducting subgroup analysis for a few included studies raises concerns about its appropriateness. The subgroup analysis lacks results for I2 and p-values due to the insufficient number of articles for the subgroup analysis. The authors should reconsider their analysis approach and provide a clear rationale for conducting subgroup analysis.

5. The references do not adhere to the journal’s guidelines. The formatting and style of the references are inconsistent. While most references are written as a list of the first author followed by “et al,” some references include the complete list of authors. The authors need to revise the references to ensure consistency and adherence to the journal's guidelines.

Reviewer #2: • In the abstract and throughout the document, rather than using prevalence of practices, better replace it with magnitude

• Abstract (results sub-section), make the factors specific. Example, department (which department)…

• Is the protocol registered in PROSPERO?

• Include data extraction sheet and JBI tool as annexes

• Data extraction and quality assessment are different. Do not mix up the statements in different section. Put the concepts in separate headings.

• Quality of the figures is poor

• in the inclusion criteria, indicate the types of study designs considered

• Outcome measurement is not clear. did you dichotomize practices of Health Science Students in preventing Hepatitis B infection? How did you do that for analysis? It has to be clear in the methods section.

6. PLOS authors have the option to publish the peer review history of their article (what does this mean?). If published, this will include your full peer review and any attached files.

Reviewer #1: **Yes: **Teshiwal Deress

Reviewer #2: **Yes: **Mesfin Gebrehiwot Damtew

---

## [Author Response · Author response to Decision Letter 0]

27 May 2024

Dear Editor(s) of PLOS ONE Journal,

We trust this message finds you well. We are pleased to submit the first revision of our manuscript titled "Practices for Preventing Hepatitis B Infection among Health Science Students in Ethiopia: A Systematic Review and Meta-Analysis" (PONE-D-24-11778).

We extend our sincere appreciation for your invaluable support throughout the review process. Your insightful comments and constructive feedback have been instrumental in refining our research. Your dedication to excellence in academic publishing is truly commendable.

We are deeply grateful for the thorough and constructive feedback provided by both the editors and reviewers. Your contributions have significantly enhanced the clarity, rigor, and overall quality of our work. The detailed suggestions and critical insights offered by the reviewers have been pivotal in addressing key areas and improving our manuscript substantially.

Thank you for considering our revised manuscript. We eagerly anticipate your feedback and remain available for any further revisions or clarifications.

Best regards,

Corresponding Author:

Gemeda Wakgari Kitil

gemedawa425@gmail.com

 

1. Point-by-point response letter to Editor

Dear Tebelay Dilnessa Academic Editor of PLOS ONE journal,

I hope this message finds you well. On behalf of my co-authors and myself, I would like to express our sincere gratitude for your invaluable efforts in handling our manuscript, "Practices for Preventing Hepatitis B Infection among Health Science Students in Ethiopia: A Systematic Review and Meta-Analysis." Your comments and suggestions have been incredibly helpful in refining our work. We greatly appreciate the time and expertise you dedicated to providing thorough and thoughtful feedback. Thank you once again for your guidance and support throughout the review process. Here is a point-by-point response addressing the changes made to your suggestions and comments:

Editor: Please submit your revised manuscript by Jul 07, 2024 11:59 PM. Please include the rebuttal letter for the response of the reviewer and editor, marked, and unmarked document when submitting your revised manuscript. 

Authors: We sincerely appreciate your insightful comment. We have taken your feedback seriously and diligently addressed the concern you mentioned. Additionally, we have ensured that all necessary documents, as per your recommendation, have been submitted along with the revised manuscript within a given time. Thank you for guiding us through this process.

Editor: When submitting your revision, we need you to address these additional requirements.

Authors: Thank you for providing us with the additional requirements for our manuscript submission to PLOS ONE. We appreciate your thorough review and guidance to ensure compliance with the journal's style and submission standards. Below, we address each of the points raised:

Editor comment-1: Please ensure that your manuscript meets PLOS ONE's style requirements, including those for file naming

Authors: Thank you for your review and specific comments. We have carefully reviewed the PLOS ONE style requirements outlined in the provided templates and ensured that our manuscript adheres to these guidelines for file naming and formatting. Any necessary adjustments have been made to align our manuscript with the specified style. Please review the revised manuscript.

Editor comment-2: Kitil., et al, assessed the practices of health science students for preventing hepatitis B infection in Ethiopia. The way presented is very poor and requires a major revision grammatically, typographically, etc.

Authors: Thank you for your detailed feedback on our manuscript titled "Practices of Health Science Students for Preventing Hepatitis B Infection in Ethiopia: Systematic Review and Meta-Analysis. We appreciate your critical insights and are committed to improving the quality of our presentation. We acknowledge that the manuscript requires significant revisions in terms of grammar, typographical errors, and overall presentation. To improve the manuscript, we have thoroughly reviewed and corrected grammatical errors, rectify typographical mistakes, and enhance overall readability. Kindly review the revised manuscript.

Editor comment-3: The title "Systematic Review and Meta-Analysis of Practices for Preventing Hepatitis B Infection among Health Science Students in Ethiopia" It is better written, ‘Practices for Preventing Hepatitis B Infection among Health Science Students in Ethiopia: Systematic Review and Meta-Analysis’.

Authors: Thank you for your valuable feedback on our manuscript title. We appreciate your suggestion and agree that the revised title enhances clarity and readability. We have updated the title accordingly. See the revised manuscript 

Editor comment-4: In the abstract part of conclusion, ‘Factors such as knowledge, age, needle stick injury exposure, and department influence these practices significantly.’ It does not give sense and needs revision. Again ‘ To improve prevention efforts, targeted interventions focusing on education and awareness are crucial. Addressing these disparities can lead to better infection control and public health outcomes in Ethiopia.’ It is a general concept and not emanated from the review result, so needs revision.

Authors: Thank you for your feedback. We have added appropriate captions for the supporting information files at the end of our manuscript and included the relevant files. We kindly invite you to review the revised manuscript, specifically Page 2, lines 40-45

Editor comment-5: In the result part of abstract and onwards, please replace the word ‘prevalence’ by ‘magnitude.’

Authors: Thank you for your invaluable comments and suggestions. We have replaced the word ‘prevalence’ with ‘magnitude’ in the abstract section as requested

Editor comment-6: In the result part of abstract ‘The statement ‘Factors significantly associated with these practices included good knowledge….’ It was vague and requires revision.

Authors: Thank you for your constructive feedback. We have revised the statement in the revised document.

Editor comment-7: In the methods part and the result part, your citations of figures were not correct. For example, (See S2 Table), (See Error! Reference source not found.), (See 9 Table 1), (see Figure 2), etc. Here, the word ‘see’ was not necessary

Authors: Thank you for bringing this to our attention. We apologize for the oversight in the citation format. We have revised the methods and results sections to ensure that the citations of figures are correctly formatted without the unnecessary use of the word 'see'. Please see the revised manuscript for further clarification.

Editor comment-8: Have you taken an ethical clearance? From whom? Is it necessary?

Authors: Thank you for your insightful suggestions and comments on ethical clearance. In response to your request, this study involves conducting a systematic review and meta-analysis, falling within the realm of secondary research where the data utilized is already published and publicly available. Given that the study strictly adheres to the Preferred Reporting Items for Systematic Reviews and Meta-Analysis (PRISMA) guidelines, which ensure transparency and rigor in reporting, ethical clearance was not deemed necessary. The primary focus of this research is to synthesize and analyze existing data, rather than directly involving human subjects or primary data collection. Therefore, ethical clearance was not sought nor required for this study.

Editor comment-9: The conclusion and recommendation were not targeted and specific

Authors: Thank you for your valuable feedback on our manuscript. We have carefully revised the conclusion and recommendation sections based on your comments to provide clearer, more focused statements. The conclusion now succinctly summarizes the key findings and their implications, and the recommendations offer actionable, specific suggestions. Please review the updated sections in our revised manuscript. See again the revised manuscript on pages 18 & 19, especially on lines 430-449.

Editor comment-10: You have to follow the manuscript writing protocol of PLOS One

Authors: Thank you for your feedback on our manuscript. We have carefully considered your comments and made the necessary revisions

Editor comment-11: Considering that the author will significantly revise the paper, major revision was recommended.

Authors: Thank you for your feedback and thorough evaluation of our work. We have carefully considered your comments and agree that significant revisions are necessary to improve the quality and clarity of the paper.

We are committed to addressing all the issues raised and undertaken a comprehensive revision to ensure that the manuscript meets the standards of the journal. We provided a detailed response to each of your suggestions and made the necessary changes accordingly.

We understand the importance of this process and assure you that we have made every effort to incorporate your feedback effectively. Thank you once again for your valuable input, and we look forward to submitting the revised manuscript in due course.

 

2. Point-by-point response letter to Reviewer-1

Authors: Thank you for your valuable feedback on our manuscript, "Practices for Preventing Hepatitis B Infection among Health Science Students in Ethiopia: A Systematic Review and Meta-Analysis." We greatly appreciate the time and effort you have invested in providing insightful comments and suggestions. We have thoroughly considered your feedback and made the necessary revisions. Below, we address each of the points raised:

Reviewer-1 Comment-1:

1. The writing lacks fluency and has serious issues with grammar that need to be addressed. 

2. The document suffers from major coherence problems and contains redundant ideas 

 Authors: Thank you for your feedback. We acknowledge the concerns regarding the lack of fluency and grammar issues in the manuscript. We have revised the structure of sentences, conducted a thorough grammar check, and sought feedback to improve the overall quality before resubmitting. Your observation regarding coherence problems and redundant ideas in the document is duly noted. We have worked on enhancing the flow of ideas and eliminating any unnecessary repetition to improve the overall clarity and coherence of the content

Reviewer-1 comment-2: The article’s searching strategy (search string) is inaccurate. The authors failed to identify the key terms related to the PICO (Population, Intervention, Comparison, Outcome) for the study. Consequently, the search string yielded an excessively large number of articles (22,150 articles), which is incorrect. With a properly structured search string, the number of articles should have been limited to several hundred. One of the main reasons for this oversight is the failure to include the country name “Ethiopia” in the search string, despite the study being conducted in Ethiopia. The authors should rectify this issue by restricting the search string with the country name.

Authors: Thank you for your feedback on our article. We appreciate your insights regarding the search strategy and its alignment with the PICO framework.

Upon careful consideration, we acknowledge the importance of refining our search strategy to better align with the PICO elements. In response to your comment, we have revised our search string to ensure the inclusion of key terms related to the Population, Intervention, Comparison, and Outcome of our study. We invited you to review our revised manuscript pages 7 & 8 lines 173-179.

Regarding your comment on the article's search strategy, we acknowledge the oversight in not including "Ethiopia" in the search string, which led to an excessive number of articles. We have rectified this by revising the search strategy to include "Ethiopia" as a key term, ensuring more focused and relevant search results aligned with the study's context. 

Reviewer-1 comment-3 On page 12, the authors stated, “The analysis, with an I2 value of 0.0% and a p-value of 0.926, revealed no substantial heterogeneity among the included studies.” However, the analysis conducted is not appropriate as it appears to be a subgroup analysis rather than a simple forest plot. Furthermore, even if the analysis was appropriate, it is unclear why the authors conducted further subgroup analysis (figure 4). Additionally, conducting subgroup analysis for a few included studies raises concerns about its appropriateness. The subgroup analysis lacks results for I2 and p-values due to the insufficient number of articles for the subgroup analysis. The authors should reconsider their analysis approach and provide a clear rationale for conducting subgroup analysis.

Authors: Thank you for bringing this to our attention. We apologize for the confusion regarding the analysis presented on page 12. There seems to have been a misunderstanding. The results mentioned in that section were derived from the pooled factors associated with preventive practice toward HBV infection, as depicted in Figure 4.

We did not conduct a subgroup analysis as indicated. The results provided were from the overall analysis of factors associated with preventive practices toward HBV infection. We understand the importance of clarity in reporting our analysis approach and rationale. We have revised the manuscript to provide a clearer explanation of the analysis conducted and the rationale behind it. Please refer to the revised tracked manuscript, page 14, line 321, for further details.

Reviewer-1 comment-4: The references do not adhere to the journal’s guidelines. The formatting and style of the references are inconsistent. While most references are written as a list of the first author followed by “et al,” some references include the complete list of authors. The authors need to revise the references to ensure consistency and adherence to the journal's guidelines.

Authors: Thank you for your feedback on our manuscript. We appreciate your attention to detail regarding the references. We apologize for the inconsistency in formatting and style. We have revised the references to ensure they adhere to the journal's guidelines consistently.

Specifically, we have ensured that all references follow the prescribed format of listing the first author followed by "et al." where applicable. Additionally, we have rectified instances where the complete list of authors was mistakenly included. For further confirmation please refer to the revised tracked References pages 21 and 22, lines 509-end, for details.

3. Point-by-point response letter to Reviewer-2

Authors: Thank you for your valuable feedback on our manuscript entitled "Practices for Preventing Hepatitis B Infection among Health Science Students in Ethiopia: A Systematic Review and Meta-Analysis." We appreciate the time you have taken to provide thoughtful comments and suggestions for improving our work. We have carefully considered your feedback and made the necessary revisions to address your comments. Below, we provide our responses to each of the points raised:

Reviewer-2 Comment-1: In the abstract and throughout the document, rather than using prevalence of practices, better replace it with magnitude.

Authors: Thank you for your valuable feedback and constructive suggestions. We have revised the abstract and the document as suggested. The term "prevalence of practices" has been replaced with "magnitude" to ensure clarity and precision throughout the manuscript. We appreciate your attention to detail and believe these changes enhance the overall quality of the paper. We invited you to review the entire revised manuscript.

Reviewer-2 Comment-2: Abstract (results sub-section), make the factors specific. Example, department (which department)…

Authors: Thank you for your insightful suggestions and feedback. We have revised the results section of the abstract to specify the departments more clearly. Please see the updated manuscript, especially the abstract and result section.

Reviewer-2 Comment-3: Is the protocol registered in PROSPERO?

Authors: Thank you for your inquiry. No, the protocol is not registered in PROSPERO. However, we have diligently adhered to stringent guidelines to guarantee transpa

---

## [Decision Letter · Decision Letter 1]

18 Jun 2024

PONE-D-24-11778R1Practices for Preventing Hepatitis B Infection among Health Science Students in Ethiopia: Systematic Review and Meta-AnalysisPLOS ONE

Dear Dr. Kitil,

Thank you for submitting your manuscript to PLOS ONE. After careful consideration, we feel that it has merit but does not fully meet PLOS ONE’s publication criteria as it currently stands. Therefore, we invite you to submit a revised version of the manuscript that addresses the points raised during the review process.

We look forward to receiving your revised manuscript.

Kind regards,

Tebelay Dilnessa, MSc

Academic Editor

PLOS ONE

Journal Requirements:

Additional Editor Comments:The paper was significantly improved, but still, it requires a proofreading for improvment of the language.Line 28: ……………………and the I2 statistic. Write as,………… and the I^2^ statistic.Please make the heading and subheading writing style uniform. For example, 3. Results; 3.1. Study selections; 3.2 Characteristics of included studies for review; 3.4 Publications bias; Funding:; Acknowledgments:, etc. The use of full stop and other punctuation marks create the confusion.In the tables’ description, the year (2024) better be at the end of the statement if the year is necessary. For example, Table 1.What is the unit/measurement of the last column of Table 1 (Practice column)Line 212-216: The subheading, ‘2.8 Patient and Public Involvement’ together with its content was not necessary. So please remove it.For the discussion; you have to use similar articles (SRMA) for compare and contrast and then justify it.Your conclusion and recommendation were too ambitious. It should be specific and based on your result.In the ‘Strengths and limitations of the study’. Select the most important strength and limitations only; avoid unnecessary concepts and details.Line 375: ‘Availability of Data and Materials:’ It is better written as, ‘Availability of data’Write properly reference number 2 based on the standard.Figure 1 is still lacking clarity; the decoration makes this table unclear. Prepare it clearly please again.

Reviewers' comments:

Reviewer's Responses to Questions

**Comments to the Author**

1. If the authors have adequately addressed your comments raised in a previous round of review and you feel that this manuscript is now acceptable for publication, you may indicate that here to bypass the “Comments to the Author” section, enter your conflict of interest statement in the “Confidential to Editor” section, and submit your "Accept" recommendation.

Reviewer #2: All comments have been addressed

2. Is the manuscript technically sound, and do the data support the conclusions?

Reviewer #2: Yes

3. Has the statistical analysis been performed appropriately and rigorously? 

Reviewer #2: Yes

4. Have the authors made all data underlying the findings in their manuscript fully available?

Reviewer #2: No

5. Is the manuscript presented in an intelligible fashion and written in standard English?

Reviewer #2: Yes

6. Review Comments to the Author

Reviewer #2: Dear authors, Thank you for considering all my comments. I believe that it can be published in its present form.

7. PLOS authors have the option to publish the peer review history of their article (what does this mean?). If published, this will include your full peer review and any attached files.

Reviewer #2: **Yes: **Mesfin Gebrehiwot

---

## [Author Response · Author response to Decision Letter 1]

20 Jun 2024

Dear Editor, Tebelay Dilnessa, MSc Academic Editor of PLOS ONE

We trust this message finds you well. We are writing to express our deepest gratitude for your invaluable support and guidance in the editorial process of our manuscript. We are pleased to submit the second revision of our manuscript, titled “Practices for Preventing Hepatitis B Infection among Health Science Students in Ethiopia: A Systematic Review and Meta-Analysis" (PONE-D-24-11778R1). We have no words to adequately thank you and truly appreciate your work.

Your detailed feedback, constructive criticism, and meticulous corrections have once again significantly enhanced the quality of the manuscript. We truly appreciate the time and effort you invested in providing such comprehensive and insightful comments. Your expertise and suggestions have not only improved the clarity and coherence of the content but also strengthened the overall argument and presentation of the study.

We are particularly grateful for your encouragement and the positive reinforcement you provided alongside your critical observations. This process has been a learning experience for us, and we are confident that the manuscript is now much more robust and impactful due to your contributions.

Thank you once again for your dedication and for helping us bring our research to its best possible form. We appreciate your consideration of our revised manuscript. We eagerly anticipate your feedback and remain available for any further revisions or clarifications.

Best regards,

Corresponding Author:

Gemeda Wakgari Kitil

gemedawa425@gmail.com

 

1. Point-by-point response letter to Editor

Dear Editor,

Once again, we greatly appreciate the time and expertise you dedicated to providing thorough and thoughtful feedback. We provide a point-by-point response addressing the changes based on your suggestions and comments as follows:

Editor: Please submit your revised manuscript by Aug 02, 2024 11:59 PM. Please include the rebuttal letter for the response of the reviewer and editor, marked, and unmarked document when submitting your revised manuscript. 

Authors: Thank you for your insightful comments. We have diligently addressed your feedback and included all required documents with the revised manuscript on time to prevent any delays.

Editor: When submitting your revision, we need you to address these additional requirements.

Authors: Thank you for providing us with the additional requirements for our manuscript submission to PLOS ONE. We appreciate your thorough review and guidance to ensure compliance with the journal's style and submission standards. Below, we address each of the points raised:

Editor comment-1: Journal Requirements:

Authors: Thank you for your instructions regarding the reference list. We have reviewed our manuscript, and no retracted articles were cited. Therefore, no changes were necessary in this regard. We appreciate your attention to this matter.

 

Editor comment-2: Line 28: ……………………and the I2 statistic. Write as,………… and the I2 statistic.

Authors: Thank you for your detailed feedback and critical insights. We are committed to improving the quality of our presentation. In response to your comment, we have reviewed the entire manuscript and replaced "I2" with “I2”. We kindly invite you to review the updated manuscript for further clarification.

Editor comment-3: Please make the heading and subheading writing style uniform. For example, 3. Results; 3.1. Study selections; 3.2 Characteristics of included studies for review; 3.4 Publications bias; Funding:; Acknowledgments:, etc. The use of full stop and other punctuation marks create the confusion.

Authors: Thank you for your feedback and suggestions regarding the uniformity of heading and subheading writing style in our manuscript. We have carefully reviewed your comments and made the necessary adjustments to ensure consistency throughout. We have addressed the issue of punctuation marks to eliminate confusion and maintain clarity across headings and subheadings. We kindly invite you to see the updated manuscript.

Editor comment-4: In the tables’ description, the year (2024) better be at the end of the statement if the year is necessary. For example, Table 1.

Authors: Thank you for your suggestion regarding the tables' descriptions. We have revised them accordingly, placing the year (2024) at the end of each statement, as recommended. Please find the updated tables in our revised manuscript. We kindly invite you to review the revised manuscript, specifically Page 10, line 241

Editor comment-5: What is the unit/measurement of the last column of Table 1 (Practice column).’

Authors: Thank you for your invaluable comments and suggestions. The unit of measurement for the last column (Practice column) in Table 1 is presented as "Percentage (%)". Each value represents the percentage of respondents or instances for each practice category. Review the updated manuscript.

Editor comment-6: Line 212-216: The subheading, ‘2.8 Patient and Public Involvement’ together with its content was not necessary. So please remove it.

Authors: Thank you for your constructive feedback. Regarding your comment on lines 212-216, we have made the necessary revisions. Specifically, we have removed the subheading "2.8 Patient and Public Involvement" as you suggested. We appreciate your thorough review and believe this change improves the clarity and structure of the manuscript.

Editor comment-7: For the discussion; you have to use similar articles (SRMA) for compare and contrast and then justify it.

Authors: Thank you for your suggestion regarding the discussion section. We have incorporated comparable systematic review and meta-analysis (SRMA) articles for contrast and justification, as per your advice. However, due to the scarcity of literature on systematic review and meta-analysis with the same topic, we prefer to discuss them alongside similar articles/primary article rather than systematic review and meta-analysis.

Editor comment-8: Your conclusion and recommendation were too ambitious. It should be specific and based on your result.

Authors: Thank you for your valuable feedback and suggestions. We have carefully reconsidered our conclusion and recommendations to ensure they are specific and grounded in our findings. We incorporated the revised conclusion and recommendations in the updated manuscript; especially on pages 15 and 16 lines 344-377.

 Editor comment-9: In the ‘Strengths and limitations of the study’. Select the most important strength and limitations only; avoid unnecessary concepts and details.

Authors: Thank you for your valuable feedback on our manuscript. Based on your comments, we have revised the Strengths and Limitations sections to focus on essential aspects while avoiding unnecessary details. Please review the updated sections in our revised manuscript, particularly on page 16, lines 391-397, for clearer, more focused statements.

 

Editor comment-10: Line 375: ‘Availability of Data and Materials:’ It is better written as, ‘Availability of data’

Authors: Thank you for your feedback on our manuscript. We have carefully considered your comments and made the necessary revisions to the updated manuscript and corrected it as “Availability of Data and Materials” was replaced with “‘Availability of data’.

 Editor comment-11: Write properly reference number 2 based on the standard.

Authors: Thank you for your feedback. In response to your Comment, we have revised reference number 2 to comply with the standard citation format. We appreciate your attention to detail and have ensured that the reference is now correctly formatted.

Editor comment-12: Figure 1 is still lacking clarity; the decoration makes this table unclear. Prepare it clearly please again.

Authors: Thank you for your feedback on Figure 1. We acknowledge that the current version lacks clarity due to excessive decoration. We have revised Figure 1 to improve its clarity by removing unnecessary decorative elements and ensuring that the information is presented in a straightforward and comprehensible manner. Please find the updated figure attached for your review.

NB. 

If there is any further information needed regarding this manuscript, please feel free to ask. We are ready to respond to your request. Additionally, we would appreciate your feedback on any revisions required or any comments that have not been fully addressed.

Thank you for your attention to this matter.

Best regards,

---

## [Decision Letter · Decision Letter 2]

24 Jun 2024

PONE-D-24-11778R2Practices for Preventing Hepatitis B Infection among Health Science Students in Ethiopia: Systematic Review and Meta-AnalysisPLOS ONE

Dear Dr. Kitil,

Thank you for submitting your manuscript to PLOS ONE. After careful consideration, we feel that it has merit but does not fully meet PLOS ONE’s publication criteria as it currently stands. Therefore, we invite you to submit a revised version of the manuscript that addresses the points raised during the review process.

We look forward to receiving your revised manuscript.

Kind regards,

Tebelay Dilnessa, MSc

Academic Editor

PLOS ONE

Journal Requirements:

Additional Editor Comments:The paper was significantly improved, again there are also more of editorial issue.Line 103: 2. Methods and materials; It is better written as, ‘**2. Materials and Methods’**Comment from previous session: Please make the heading and subheading writing style uniform. For example, 3. Results; 3.1. Study selections; 3.2 Characteristics of included studies for review; 3.4 Publications bias, etc. The use of full stop and other punctuation marks create the confusion.In the recommendation part remove the subheadings (Education and Awareness Programs, Training on Safe Practices, Curriculum Integration, Support Services and Policy DevelopmentLine 378: Declaration of Non-Competing Interests: First avoid the use of colon mark (:).  It is better written as, Competing interests: The authors have declared that no competing interests exist.The order of numbering better be removed from (6-8), but leave reference numbering and make it; ‘**6. References’**You can remove the ‘Authors’ contribution’ as the system creates it.Please revise all the references. For example, references such as 11, 12, 14, 17, 19, 20, 21, 23, 24 written correctly as followsAynalem A, Deribe B, Ayalew M, Mamuye A, Israel E, Mebratu A, *et al*. Practice towards Hepatitis B Virus Infection Prevention and Its Associated Factors among Undergraduate Students at Hawassa University College of Medicine and Health Sciences, Hawassa, Sidama, Ethiopia, 2021: Cross-Sectional Study. *Int J Hepatol*. 2022; 2022:2673740. doi: 10.1155/2022/2673740. PMID: 35991003; PMCID: PMC9391155.Noubiap JJ, Nansseu JR, Kengne KK, Tchokfe Ndoula S, Agyingi LA. Occupational exposure to blood, hepatitis B vaccine knowledge and uptake among medical students in Cameroon. *BMC Med Educ*. 2013; 13:148. doi: 10.1186/1472-6920-13-148.Demsiss W, Seid A, Fiseha T. Hepatitis B and C: Seroprevalence, knowledge, practice and associated factors among medicine and health science students in Northeast Ethiopia. *PLoS One.* 2018;13(5): e0196539. doi: 10.1371/journal.pone.0196539.Hutin Y, Hauri A, Chiarello L, Catlin M, Stilwell B, Ghebrehiwet T, *et al*. Injection Safety Best Practices Development Group. Best infection control practices for intradermal, subcutaneous, and intramuscular needle injections. *Bull World Health Organ*. 2003;81(7):491-500.Mesfin YM, Kibret KT. Assessment of knowledge and practice towards hepatitis B among medical and health science students in Haramaya University, Ethiopia. *PLoS One*. 2013; 8(11): e79642. doi: 10.1371/journal.pone.0079642.Moher D, Liberati A, Tetzlaff J, Altman DG; PRISMA Group. Preferred reporting items for systematic reviews and meta-analyses: the PRISMA statement. *PLoS Med*. 2009; 6(7): e1000097. doi: 10.1371/journal.pmed.1000097.Awoke N, Mulgeta H, Lolaso T, Tekalign T, Samuel S, Obsa MS, *et al*. Full-dose hepatitis B virus vaccination coverage and associated factors among health care workers in Ethiopia: A systematic review and meta-analysis. *PLoS One*. 2020;15(10): e0241226. doi: 10.1371/journal.pone.0241226. Haile K, Timerga A, Mose A, Mekonnen Z. Hepatitis B vaccination status and associated factors among students of medicine and health sciences in Wolkite University, Southwest Ethiopia: A cross-sectional study. *PLoS One*. 2021;16(9): e0257621. doi: 10.1371/journal.pone.0257621.Rachiotis G, Goritsas C, Alikakou V, Ferti A, Roumeliotou A. Vaccination against hepatitis B virus in workers of a general hospital in Athens. *Med Lav*. 2005; 96(1):80-6.

Reviewers' comments:

Reviewer's Responses to Questions

**Comments to the Author**

1. If the authors have adequately addressed your comments raised in a previous round of review and you feel that this manuscript is now acceptable for publication, you may indicate that here to bypass the “Comments to the Author” section, enter your conflict of interest statement in the “Confidential to Editor” section, and submit your "Accept" recommendation.

Reviewer #2: All comments have been addressed

2. Is the manuscript technically sound, and do the data support the conclusions?

Reviewer #2: Yes

3. Has the statistical analysis been performed appropriately and rigorously? 

Reviewer #2: Yes

4. Have the authors made all data underlying the findings in their manuscript fully available?

Reviewer #2: Yes

5. Is the manuscript presented in an intelligible fashion and written in standard English?

Reviewer #2: Yes

6. Review Comments to the Author

Reviewer #2: Dear authors, Thank you for considering all my comments. I believe that it can be published in its present form.

7. PLOS authors have the option to publish the peer review history of their article (what does this mean?). If published, this will include your full peer review and any attached files.

Reviewer #2: **Yes: **Mesfin Gebrehiwot

---

## [Author Response · Author response to Decision Letter 2]

25 Jun 2024

Dear Editor, Tebelay Dilnessa, 

Academic Editor of PLOS ONE

We are truly grateful for your unwavering support and guidance throughout the editorial journey of our manuscript titled “Practices for Preventing Hepatitis B Infection among Health Science Students in Ethiopia: A Systematic Review and Meta-Analysis" (PONE-D-24-11778R2). It is with great pleasure that we submit the third revised version of our manuscript to you.

Your meticulous feedback, constructive criticism, and insightful corrections have not only refined the manuscript but also enriched its depth and impact. Your expertise has played a crucial role in improving the clarity and coherence of our study, and we deeply appreciate the time and effort you have dedicated to this process.

We are particularly grateful for your encouragement and positive reinforcement alongside your thorough critiques. This collaborative effort has been an enlightening experience for us, and we are confident that the manuscript is now stronger and more compelling thanks to your invaluable contributions.

Thank you once again for your dedication in helping us shape our research into its best possible form. We eagerly await your feedback on this revised submission and remain available for any further adjustments or clarifications you may require.

Warm regards,

Corresponding Author: Gemeda Wakgari Kitil

gemedawa425@gmail.com

 

1. Point-by-point response letter to Editor

Dear Editor,

We sincerely appreciate the thorough and thoughtful feedback you've provided once again. In response to your suggestions, we have meticulously revised our manuscript. Below is our detailed response addressing the changes:

Editorial Deadline: Please submit your revised manuscript by Aug 08, 2024, 11:59 PM, including the rebuttal letter and both marked and unmarked document versions.

Authors: Thank you for your insightful comments. We have promptly incorporated all necessary documents with our revised submission to ensure timely processing.

Editor comment-1: Journal Requirements:

Authors: Thank you for your guidance on the reference list requirements. We have thoroughly reviewed our manuscript and confirmed that no retracted articles were cited. Therefore, no changes were necessary in this regard. We appreciate your attention to this matter.

Editor comment-2: Line 103: 2. Methods and materials; It is better written as, ‘2. Materials and Methods’.

Authors: Thank you for your valuable feedback. We have revised the section title from "2. Methods and materials" to "2. Materials and Methods" to adhere to conventional scientific writing standards. This change should enhance the clarity and readability of the document. We kindly invite you to review the updated manuscript on page 5 for further clarification.

Editor comment-3: Comment from previous session: Please make the heading and subheading writing style uniform. For example, 3. Results; 3.1. Study selections; 3.2 Characteristics of included studies for review; 3.4 Publications bias, etc. The use of full stop and other punctuation marks create the confusion.

Authors: Thank you for your feedback. We would like to say sorry for the previously unaddressed comment. We have reviewed the document and made the necessary adjustments to ensure consistency in the headings and subheadings. They now follow a uniform style without unnecessary punctuation marks, such as full stops, which should enhance readability and clarity throughout the manuscript.

Editor comment-4: In the recommendation part remove the subheadings (Education and Awareness Programs, Training on Safe Practices, Curriculum Integration, Support Services and Policy Development.

Authors: Thank you for your suggestion. We have revised the recommendation section by removing the subheadings (Education and Awareness Programs, Training on Safe Practices, Curriculum Integration, Support Services, and Policy Development). This change aims to streamline the presentation and ensure a more cohesive flow of recommendations. We kindly invite you to review the revised manuscript, specifically Page 15, lines 346-362

Editor comment-5: Line 378: Declaration of Non-Competing Interests: First avoid the use of colon mark (:). It is better written as, Competing interests: The authors have declared that no competing interests exist.

Authors: Thank you for your comment. We have revised the statement as follows: "Competing interests: The authors have declared that no competing interests exist." This adjustment removes the colon and maintains clarity regarding our declaration of non-competing interests. We invite you to review the updated manuscript.

Editor comment-6: The order of numbering better be removed from (6-8), but leave reference numbering and make it; ‘6. References’.

Authors: Thank you for your constructive comments and suggestions. We have removed the numbering from (6-8) and retained the reference numbering, now titled “6. References”. This adjustment aims to improve clarity and consistency in the document structure. We kindly invite you to see the revised manuscript.

Editor comment-7: You can remove the ‘Authors’ contribution’ as the system creates it.

Authors: Thank you for your input. We've removed the 'Authors' contribution' section as it is automatically generated by the system, ensuring a more streamlined presentation of the document.

Editor comment-8: Please revise all the references. For example, references such as 11, 12, 14, 17, 19, 20, 21, 23, 24 are written correctly as follows.

Authors: Thank you for highlighting the need for revisions to our references. We have carefully reviewed and corrected all references as per your instructions. The revised manuscript now accurately reflects the correct formatting for each cited source. In the tracked manuscript version, references 23, 26, 27, and 31 were highlighted in red color. These references were duplicates of the revised references and have been removed in the cleaned manuscript.

---

## [Editor Report · Decision Letter 3]

26 Jun 2024

Practices for Preventing Hepatitis B Infection among Health Science Students in Ethiopia: Systematic Review and Meta-Analysis

PONE-D-24-11778R3

Dear Dr. Kitil,

We’re pleased to inform you that your manuscript has been judged scientifically suitable for publication and will be formally accepted for publication once it meets all outstanding technical requirements.

Kind regards,

Tebelay Dilnessa, MSc

Academic Editor

PLOS ONE
---

## [Editor Report · Acceptance letter]

28 Jun 2024

PONE-D-24-11778R3 

PLOS ONE

Dear Dr. Kitil, 

I'm pleased to inform you that your manuscript has been deemed suitable for publication in PLOS ONE. Congratulations! Your manuscript is now being handed over to our production team.

Kind regards, 

on behalf of

Dr. Tebelay Dilnessa 

Academic Editor

PLOS ONE